# Dual Stimuli-Responsive Multifunctional Silicon Nanocarriers for Specifically Targeting Mitochondria in Human Cancer Cells

**DOI:** 10.3390/pharmaceutics14040858

**Published:** 2022-04-13

**Authors:** Vy Anh Tran, Giau Van Vo, Mario A. Tan, Joon-Seo Park, Seong Soo A. An, Sang-Wha Lee

**Affiliations:** 1Department of Chemical and Biological Engineering, Gachon University, 1342 Seongnam-daero, Sujung-gu, Seongnam-si 461-701, Gyeonggi-do, Korea; tranhvy@gmail.com; 2Department of Biomedical Engineering, School of Medicine, Vietnam National University Ho Chi Minh City (VNU-HCM), Ho Chi Minh City 700000, Vietnam; giauvvo@gmail.com; 3Vietnam National University Ho Chi Minh City (VNU-HCM), Ho Chi Minh City 700000, Vietnam; 4College of Science and Research Center for the Natural and Applied Sciences, University of Santo Tomas, Manila 1015, Philippines; matan@ust.edu.ph; 5Department of Chemistry, Eastern University, 1300 Eagle Road, St. Davids, PA 19087, USA; jpark6@eastern.edu; 6Department of Bionano Technology, Bionano Research Institute, Gachon University, 1342 Seongnam-daero, Sujung-gu, Seongnam-si 461-701, Gyeonggi-do, Korea

**Keywords:** mesoporous silicon, controlled drug release, graphene oxide, dequalinium, mitochondria targeting

## Abstract

Specific targeting, selective stimuli-responsiveness, and controlled release of anticancer agents are requested for high therapeutic efficiency with a minimal adverse effect. Herein, we report the sophisticated synthesis and functionalization of fluorescent mesoporous silicon (FMPSi) nanoparticles decorated with graphene oxide (GO) nanosheets. GO-wrapped FMPSi (FMPSi@GO) was loaded with a cisplatin (Cis) anticancer agent, and Cis-loaded FMPSi@GO (FMPSi-Cis@GO) exhibited the dual stimuli (pH and NIR)-responsiveness of controlled drug release, i.e., the drug release rate was distinctly enhanced at acidic pH 5.5 than at neutral pH 7.0 and further enhanced under NIR irradiation at acidic pH condition. Notably, dequalinium-conjugated FMPSi-Cis@GO (FMPSi-Cis@GO@DQA) demonstrated an excellent specificity for mitochondrial targeting in cancer cells without noticeable toxicity to normal human cells. Our novel silicon nanocarriers demonstrated not only stimuli (pH and NIR)-responsive controlled drug release, but also selective accumulation in the mitochondria of cancer cells and destroying them.

## 1. Introduction

Mitochondria are double-membrane-bound organelles for producing cellular ATP, regulating intracellular calcium homeostasis, generating reactive oxygen species, activating the intrinsic apoptotic pathway, and producing hormones; therefore, aberrations in mitochondrial physiology are involved in various diseases [1,2]. Cancer cells predominantly produce chemical energy via a high rate of glycolysis, even in the presence of abundant oxygen. Therefore, this organelle is increasingly paid attention as a “prime target” for pharmacological intervention [3,4,5]. However, it is difficult for drug molecules to enter the mitochondria due to the extensively folded and compartmentalized structure of the inner mitochondrial membrane. The mitochondrial membrane potential of cancer cells is highly negative, which is 3- to 5-fold higher than that of the plasma membrane. In this regard, positively charged target molecules can easily accumulate in the mitochondria organelle [6]. Several potential mitochondrial delivery systems have been developed using specific biochemical agents such as triphenylphosphonium (TPP), mitochondria penetrating peptides (MPPs), and mitochondria targeting signal peptides (MTS) [7]. Mitochondria-targeting nanoparticles (NPs) should be developed to deliver the therapeutic agents to mitochondria without significant drug resistance. As a result, they are localized in mitochondria to damage the mitochondrial membrane, consequently inducing greater cytotoxicity and apoptotic cell death [6].

Among current materials such as silica [8,9], metal organic frameworks [10,11,12,13,14,15], composites [16,17,18,19], and metal compounds [20,21,22], mesoporous silicon NPs have received considerable attention in the field of biomedical drug delivery due to their outstanding properties such as in vitro biocompatibility, in vivo biodegradability, and high photothermal activity [23,24]. A complete magnesiothermic reduction of mesoporous silica NPs (MSN) has provided a platform for the rational design of multifunctional silicon nanostructures for controlled and targeted delivery of therapeutic agents [25,26]. In the following literature studies, anticancer drugs such as paclitaxel and doxorubicin were loaded into porous silicon (PSi) NPs coated with Pluronic F-127, PEG, and hyaluronic acid layers, but poor targeting to specific organs limited their applications to cancer therapy [27,28]. Alginate and chitosan were also coated on PSi with targeting ligands for the construction of multi-responsive nanocarriers, but the PSi composites showed poor drug-capturing ability due to their low porosity, resulting in inefficient cancer cell killing activity [24]. Undecylenic acid functionalized-PSi NPs were conjugated with β-cyclodextrin to study the impact of surface polymeric functionalization on the physical and biological properties of drug-loaded PSi, based on their anticancer effects on MCF-7 and MDA-MB-231 breast cancer cells [29,30].

To enhance anti-cancer efficacy, various functional strategies of multiple integrations are highly required [31,32]. The smart drug delivery systems (DDS) should possess cancer cell organelle-targeting specificity, bio-imaging, and therapeutic activity to achieve excellent anticancer effects on the apoptotic death of cancer cells [33]. As one of the smart DDS materials, graphene oxide (GO) has multiple hydrophilic groups that can facilitate drug adsorption on its surface and edges via electrostatic, hydrogen bonding, and π–π stacking interactions [19,34,35]. The hydrophilic groups (e.g., COOH and OH) of GO improve the solubility in the aqueous phase and π conjugations contribute to the strong NIR absorption of GO [36,37,38]. In this context, GO nanosheets conjugated with silicon-based nanocarriers can provide strong anticancer efficacy, chemo-photothermal sensitivity, and stimuli-responsive drug release [36,39]. Furthermore, GO can provide reactive sites for surface modification and excellent biocompatibility, which are required for targeted drug delivery [40].

Herein, we first synthesized mesoporous silicon (MPSi) NPs grafted with fluorescent organic conjugates, so-called fluorescent MPSi (FMPSi), which were subsequently decorated with GO nanosheets. The GO-wrapped FMPSi (FMPSi@GO) loaded with cisplatin drug exhibited the dual stimuli (pH and NIR)-responsive controlled drug release, i.e., the drug release rate was increased by pH lowering (from pH 7.4 to pH 5.5) and significantly increased at acidic pH condition under NIR irradiation. Furthermore, dequalinium (DQA)-conjugated FMPSi@GO, FMPSi@GO@DQA, provided the nanocarrier with remarkable mitochondria targeting specificity, consequently leading to a significant decrease in cellular ATP production and damaging mitochondrial membranes of cancer cells. The FITC fluorescence of isolated mitochondria from treated cells (HeLa, SH-SY5Y, and HEK293) was detected using confocal microscopy to track the mitochondria binding capability of the NPs. Cross-sectional TEM images were obtained to provide direct evidence for the internalization of NPs in the mitochondria of cancer cells.

## 2. Materials and Methods

### 2.1. Chemical Materials

The following chemicals were purchased from Sigma-Aldrich (Seoul, Korea) and were used without further purification: 3-aminopropyltrimethoxysilane (APTMS, 97%), ammonium fluoride (NH_4_F, 99.99%), cetyltrimethylammonium bromide (CTAB, 99%), cisplatin, dequalinium (DQA, ≥95%), dimethyl sulfoxide (DMSO, ≥99.9%), powdered dopamine hydrochloride, fluorescein isothiocyanate (FITC, 90%), graphene oxide (GO, 4 mg/mL), magnesium (Mg, 20–230 mesh, 98%), *N*-(3-dimethylaminopropyl)-*N*′-ethylcarbodiimide hydrochloride (EDC, crystalline), *N*-hydroxysuccinimide (NHS, 98%), sodium hydroxide (NaOH, ≥98%), and tetraethyl orthosilicate (TEOS, 99%).

Phosphate-buffered saline (PBS) was obtained from Bioneer (Seoul, Korea). Commercially available absolute ethanol (EtOH) and HPLC-grade H_2_O were used without further purification. An acidic solution of HNO_3_:HCl (1:3 *v*/*v*) was used to clean the glassware followed by deionized (DI) H_2_O.

### 2.2. Synthesis of Nanocarriers

#### 2.2.1. Synthesis of Mesoporous Silica Nanoparticles

Mesoporous silica nanoparticles (MSN) were synthesized using the cationic surfactant CTAB via a sol–gel reaction with TEOS. Briefly, 0.30 g of CTAB and 0.80 g of NH_4_F were mixed in 150 mL of deionized (DI) water and heated to 80 °C with vigorous stirring. When they were dissolved completely, 3 mL of TEOS was added dropwise and mixed to achieve a milky white solution at 80 °C for 12 h. To obtain the white solid of MSN, the milky solution was centrifuged (8000 rpm, 10 min), followed by washing with DI H_2_O and EtOH several times, and freeze-drying for 24 h. To allow the removal of the surfactant template (CTAB), the white precipitate was dissolved in 150 mL of EtOH containing 2.0 mL of HCl and refluxed for 24 h at 80 °C. This step was repeated to ensure the complete removal of the surfactant. The obtained product was centrifuged, washed with DI water, and freeze-dried for 24 h [8,41].

#### 2.2.2. Magnesiothermic Reduction of MSN into MPSi

A 500 mg of MSN and 450 mg of magnesium powder (325 mesh) were mixed and loaded into SS 316 Swagelok-type reactors filled with argon atmosphere in the glove box. The SS 316 Swagelok-type reactor was heated in a tube furnace at 675 °C for 5 h (5 °C/min) in an argon atmosphere and then cooled to room temperature (rt). The products were immersed in 10 mL of HCl (2 M) for 7 h to remove MgO, Mg_2_Si, and undesired products, followed by three times washes with DI water. The products were washed with copious amounts of HF (5 wt%) solution to remove the residual silica. The powder was rinsed several times with DI water, dispersed in ethanol, and finally dried using the freeze dryer for 48 h [25,42,43].

### 2.3. Functionalization of Fluorescent Conjugates onto MPSi

Prior to surface functionalization, the MPSi samples were treated in a furnace at 100 °C for 1 h and then dispersed in DI water. This treatment increases the number of hydroxyl groups on the surface of MPSi (MPSi-OH). Next, the hydrolyzed APTMS solution was obtained by mixing 19 mL of ethanol and 5 mL of DI water, followed by adding an aliquot of APTMS solution dropwise under continuous stirring. This reaction resulted in the chemical modification of methoxy groups of APTMS ((OCH_3_)_3_-Si-(CH_2_)_3_-NH_2_) into hydroxyl groups of silanetriols ((OH)_3_-Si-(CH_2_)_3_-NH_2_). Subsequently, MPSi-OH NPs were incubated in the freshly hydrolyzed APTMS solution for 12 h under constant stirring, and the product samples were rinsed with anhydrous ethanol to remove any unbound APTMS-derivatives and dried in the freeze dryer. This treatment led to the grafting of hydrolyzed APTMS groups onto the pore walls of MPSi, and the grafted MPSi (MPSi@APTMS) was progressively conjugated with other functional molecules [44,45].

FITC-conjugated APTMS (APTMS-FITC) was prepared by adding MPSi@APTMS into FITC in ethanol (1.25 mg/mL). To obtain APTMS-FITC-grafted MPSi (MPSi@APTMS-FITC), i.e., fluorescent MPSi (so-called FMPSi), the mixture was shaken at room temperature (rt ~20 °C) in the dark for 12 h. The product was washed with ethanol, centrifuged to remove excess FITC, and dried in the freeze dryer [8].

### 2.4. Drug Loadings and In Vitro Release Tests

#### 2.4.1. Loading Cisplatin into FMPSi

Cisplatin is a weakly acidic drug with a pK_a_ from 5.5 to 7.3 in its monohydrated and dihydrated complex form [46]. Cisplatin solubility in DI water is 2.5 mg/mL (at 25 °C), which increases up to 4 mg/mL when the temperature is increased up to 35 °C. The solubility of cisplatin is notably increased in DMSO solvent (25 mg/mL) at 25 °C [47]. Cisplatin is sensitive to light, and all the synthesis steps were performed without exposure to light. For drug loading, 150 mg of the FMPSi sample was dispersed in 5 mL of DMSO containing 50 mg of cisplatin. The mixed solution was stirred at rt for 24 h to load maximum amounts of cisplatin. The cisplatin-loaded FMPSi (FMPSi-Cis) was separated by centrifugation at 7000 rpm for 7 min. The supernatant was collected to determine the loading amount of cisplatin by calculating the difference between the initial and residual amounts of cisplatin in the solution [48].

#### 2.4.2. Wrapping with GO Layers and Conjugation with QDA

GO (0.1 g; 0.025 mL of GO suspension) was sonicated in 2 mL of DI water for 3 min. The FMPSi-Cis (50 mg) and GO solution were mixed and dispersed in DI water using ultrasonic vibration for 3 min. After ultrasonic dispersion, the mixed solution was stirred at 60 °C for 2 h. The precipitate was washed with DI water several times. Then, the final product (FMPSi-Cis@GO) was collected by centrifugation and dried in the freeze dryer system.

DQA is well-known for its mitochondria-targeting specificity. DQA-conjugated FMPSi-Cis@GO (FMPSi-Cis@GO@DQA) was prepared using the coupling reaction between the amine groups of DQA and the carboxyl group of GO using EDC-NHS agents [36]. Briefly, 5 mg of DQA was dissolved in 5 mL of DI water. To activate the carboxylic acid groups of GO, EDC (20 μL of 1 mg/mL) was added to the FMPSi-Cis@GO solution (50 mg of NPs in 20 mL of DI water), then 50 μL of NHS (1 mg/mL) was added under continuous stirring. The DQA solution was added dropwise into the solution. The reaction mixture was gently stirred with a magnetic stirrer at rt for 2 h. Then, the final product was centrifuged and washed with DI water several times to remove the residual impurities [49].

#### 2.4.3. In Vitro Ph/Nir Irradiation-Controlled Drug Release

The FMPSi-Cis@GO were dissolved in 10 mL of PBS under constant stirring at 37 ± 1 °C. In vitro drug release test started immediately after the MPSi samples were added to the PBS solution. The solution was periodically sampled to measure the absorbance changes of the released cisplatin in the PBS solution at 301 nm by ultraviolet–visible (UV–vis) spectroscopy (NanoDrop; NanoDrop Technologies, Wilmington, DE, USA) at the Smart Materials Research Center for IoT at Gachon University. The absorbance values were converted into the released amounts of cisplatin using the standard curve based on the linear correlation between absorbance and corresponding concentration.

To investigate the stimuli (pH and NIR irradiation)-responsive drug release behavior of as-prepared samples, in vitro release tests were carried out in 10 mL of PBS solutions at different pHs (pH 5.5 and pH 7.4) using FMPSi-Cis@GO with (or without) NIR irradiation at 808 nm (1.0 W/cm^2^) for 20 min. The NIR laser was periodically employed for 20 min irradiation at the release times of 0, 5, 10, 20, and 30 h. During the continuous release process, the solution was periodically sampled and the absorbance was measured at 301 nm using UV–vis spectroscopy. Finally, the released amounts of drug were calculated based on the linear correlation of the standard curve, and the fractional release of the drug was plotted versus corresponding times.

### 2.5. Cell Cytotoxicity Assay Using the MPSi-Drug System

#### 2.5.1. Cell Viability Assay

Human cervical cancer (HeLa, American Type Culture Collection (ATCC CCL-2), Manassas, VA, USA), neuroblastoma (SH-SY5Y, ATCC CRL-2266), and embryonic kidney (HEK293T, ATCC CRL-1573) cell lines were incubated in Dulbecco’s modified Eagle’s medium (DMEM) with 10% FBS and 1% antibiotics (penicillin-streptomycin, 10,000 U/mL) at 37 °C in a humidified atmosphere containing 5% CO_2_. Briefly, different (HeLa, SH-SY5Y, and HEK293) cell lines were cultured in 96-well plates with a density of 2 × 10^4^ cells per well. After incubation for 24 h, the old medium was replaced with fresh medium containing different NP samples of various concentrations (100, 50, 25, 12.5, 6.5, 3.3, and 1.3 μg/mL). After another 48 h of incubation, the old medium was removed, followed by washing thrice with PBS, followed by the addition of 100 μL of fresh medium to each well. After incubation for 30 min, CellTiter-Glo Luminescent reagent (100 μL) (Promega, Madison, WI, USA) was added, followed by gentle shaking for 10 min at rt. The luminescence signal was measured using a microplate reader (Perkin Elmer, Victor X5, Waltham, MA, USA). The percentage cell viability was calculated based on the control (untreated) cells. The values were expressed as mean ± SD of triplicate experiments.

#### 2.5.2. Cellular Uptake and Intracellular Distribution

SH-SY5Y cells were seeded in full media to a final density of a 1 × 10^5^ cells/well in 6-well plates in the presence of sterile rounded 22 × 1.5 mm glass coverslips. The next day, the cells were treated with 25 μg/mL FMPSi, FMPSi-Cis, FMPSi-Cis@GO, and FMPSi-Cis@GO@DQA for 4 h, 8 h, or 12 h. Next, each slide was washed thrice with PBS. For staining the mitochondria, media was replaced with fresh media containing Mito-tracker Deep red (Thermo Scientific, Waltham, MA, USA) staining solution (50 nM) and cultured in the incubator for an additional 45 min. The cells were then washed with PBS (3×) and fixed with 4% paraformaldehyde in PBS for 10 min at rt. The cells fixed on the slides were washed with PBS (3×) followed by staining with DAPI in PBS (1 μg/mL) for 20 min at rt. The slides were mounted with Eukitt^®^ Quick-hardening mounting medium (Sigma-Aldrich, St. Louis, MO, USA) for visualization by confocal microscopy. The mounted cells were imaged at λ = 404, 488, and 638 nm to measure the fluorescence intensity of DAPI, FITC, and Mito-Tracker Deep, respectively, using confocal microscopy (Nikon Eclipse TE2000-S, C1 Plus; Nikon, Tokyo, Japan)

#### 2.5.3. Mitochondria Isolation and Analysis of Fluorescence Intensity

To further validate the successful targeting of mitochondria in cancer cells, mitochondrial isolation was conducted in HeLa cells, following the instructions of the mitochondria isolation kit (Thermo Scientific). HeLa cells were cultured in a T-75 cell culture flask at a density of 1 × 10^7^ cells/mL. Next, old media was replaced by fresh media containing 15 µg/mL of different FMPSi-Cis@GO and FMPSi-Cis@GO@DQA samples, and the cells were further incubated for 24 h. Subsequently, old media was removed, and the cells were washed thrice with PBS. The adherent cells cultured in a T-75 cell culture flask were trypsinized, the cells were pelleted by centrifuging the cell suspension in a 2.0 mL centrifuge tube at 850× *g* for 2 min, and the supernatant was carefully discarded. The pelleted cells were treated with a cell rupturing reagent A (800 μL) by vortexing at medium speed for 5 s, followed by ice water incubation for 2 min. Then, the mitochondrial isolation reagent B (10 μL) was added, and the cell mixture was vortexed for 5 s. After ice incubation for 5 min, 800 μL isolation reagent C was added to the cell-containing tube, and the mixture gently inverted several times. The mixture was then centrifuged for 10 min at 700× *g*, and the supernatant was collected in a new 2.0 mL tube. The isolated mitochondria were pelleted by centrifuging the tube at 12,000× *g* at 4 °C for 15 min. Subsequently, the isolated mitochondria were resuspended in isolation reagent C and re-centrifuged twice to eliminate the free nanoparticles. Then, the mitochondria-containing pellets were suspended in PBS (200 μL) and transferred to a 96-well plate. The fluorescence intensity at an excitation wavelength of 488 nm (FITC) was measured using a microplate reader (Perkin Elmer, Victor X5, Waltham, MA, USA).

### 2.6. Data Analysis

Cell viability values were reported as the mean ± standard deviations (SD) of triplicate experiments. The 50% inhibitive concentrations (IC_50_) of the NPs were calculated using a nonlinear regression curve fit (GraphPad Prism ver. 6, San Diego, CA, USA).

## 3. Results and Discussion

### 3.1. Synthesis and Characterization of Multifunctional Mesoporous Silicon (MPSi) NPs

MPSi samples were pre-treated in a furnace for 1 h at 100 °C and then dispersed in water (Figure 1). After thermal treatment to increase the number of surface OH groups, the MPSi-OH was functionalized with 3-aminopropyltrimethoxysilane (APTMS), and the resulting APTMS-MPSi was conjugated with fluorescein isothiocyanate (FITC), forming fluorescent APTMS-FITC-conjugated MPSi (hereafter, APTMS-FITC will be abbreviated as AP-FI). The fluorescent MPSi (FMPSi) can load high amounts of drug molecules due to the increased surface area. Notably, the AP-FI conjugates can provide not only tortuous pathways through the porous channels of MPSi, but also control the release of drug molecules via electrostatic and/or hydrogen bonding interactions [8]. To test the anticancer efficacy of FMPSi-based nanocarriers, Cisplatin (Cis)-loaded FMPSi (FMPSi-Cis) was coated with GO nanosheets, forming the FMPSi-Cis@GO. GO wrapping can enhance the efficiency of photothermal therapy and cellular uptake of FMPSi [50,51]. To improve the mitochondrial selectivity and cancer cell toxicity, DQA was further conjugated with FMPSi-Cis@GO to develop mitochondria-targeted nanocarriers labelled as FMPSi-Cis@GO@DQA [52], which can transport chemotherapeutics to mammalian mitochondria in living cancer cells [53,54].

To verify the successful surface modifications during the consecutive synthesis of multifunctional NPs, the zeta potential of the stepwise product was determined at neutral pH 7.4, which is summarized in Appendix A. The zeta potential of MPSi was measured to be +0.90 mV, but the sign of zeta-potential was inverted to a negative value of −8.16 mV after heat treatment at 100 °C, indicating the formation of hydroxyl groups on the surface. However, the zeta potential of MPSi was slightly changed into a positive value of +3.85 mV after the surface grafting of hydrolyzed APTMS with terminated amine groups. The subsequent conjugation of fluorescent FITC with APTMS, i.e., forming AP-FI conjugates, changed the zeta potential into a negative value of −14.95 mV, confirming the replacement of positive amine groups with negative carboxylic acid groups, as illustrated in Figure 1. Finally, GO wrapping on the FMPSi further decreased the zeta potential to −34.80 mV. For reference, pK_a_ values of GO are known to be 4.3, 6.6, and 9.8, so GO have multiple negative charges at neutral pH conditions [55].

Figure 1A illustrates the FTIR spectra of the samples MPSi, FMPSi, and FMPSi@GO. The MPSi showed FTIR peaks at 802 cm^−1^ and 1294 cm^−1^, correlating to the Si-Si and Si-O-Si bonds, respectively. After binding with AP-FI conjugates, the FMPSi exhibited a new peak at 1585 cm^−1^ that was attributed to the C=S vibration. Moreover, the broad bands at 1404 cm^−1^, 2314 cm^−1^, and 3662 cm^−1^ were assigned to C-N, N-C-N, and NH groups of AP-FI conjugates, respectively. The FMPSi@GO exhibited three new peaks at 3550 cm^−1^, 2950 cm^−1^, and 2337 cm^−1^, which were attributed to the O-H, C-H, and COO groups originating from wrapped GO nanosheets, respectively. Figure 1B shows the Raman spectra of the samples. MPSi (or FMPSi) revealed a distinct Raman peak at ~500 cm^−1^, indicating the successful transformation of Si nanocrystals from silicon oxide by the Mg reduction process [56]. The Raman spectra of FMPSi@GO showed two broad bands at 1359 cm^−1^ (D band) and 1591 cm^−1^ (G band) that matched with the Raman bands of pure GO [57].

Figure 1C shows the BET surface area and porosity of the samples MSN, MPSi, and FMPSi. The N_2_ adsorption/desorption isotherm curve of MSN was similar to type III isotherm, which did not exhibit any limiting adsorption at the high ratio of p/p_0_ [58]. An increase in the adsorption capacity at the initial stage of adsorption isotherm manifested an abundance of nanopores [36]. Moreover, the observed sharpness in the pressure range of 0.85–1.0 indicated the distribution of narrow pore size [8,59]. According to Barrett–Joyner–Halenda (BJH) analysis (Appendix A), MSN has a mesoporous structure with a narrow pore size distribution of ~2.78 nm. After the magnesiothermic reduction of MSN, the resulting MPSi exhibited a hybrid type III/IV isotherm, displaying a characteristic desorption branch associated with the hysteresis loop closure [60]. The MPSi showed a slightly narrow pore size distribution of ~2.52 nm, indicative of distortion of the mesostructured MPSi by Mg vapor-mediated de-oxidation of Si-O-Si bonds. After grafting with fluorescent AP-FI conjugates, the FMPSi presented a typical type III curve representing the enlarged mesopores, showing the pore size distribution of ~3.92 nm [58]. The specific surface areas were measured as 1407 m^2^g^−1^ and 427 m^2^g^−1^ for MSN and MPSi samples, respectively (Appendix A). The MPSi has a relatively low surface area, probably due to the adventitious pore-blocking by magnesium-reductive calcination effect. However, the FMPSi exhibited an increased surface area of 807 m^2^g^−1^ because the pore size of FMPSi was increased by pore wetting phenomena of grafted AP-FI conjugates [61]. It is expected that the highly porous structure of FMPSi is beneficial for accommodating large amounts of anticancer drugs as a controlled drug delivery system.

Figure 1D shows TEM images that depicted the structural evolution starting from MSN to MPSi and FMPSi@GO. The TEM image of MSN (Figure 1D1) showed an average particle size of ~100 nm and indicated the presence of uniformly distributed mesopores inside the MSN [62]. The abundance of these mesopores is advantageous for the rapid diffusion of Mg vapor and facilitated the dissipation of reaction heat, preventing excessive fusion of reduced Si nanocrystals [63]. Judging from the overall morphologies and particle sizes, the spherical shape of MPSi was preserved during the magnesium reduction process of MSN. Figure 1D2 shows the TEM images of MPSi grafted with AP-FI conjugates. Figure 1D3 shows the TEM image of GO-wrapped FMPSi, showing the presence of ultrathin wrapping of GO nanosheets. Figure 1D4 indicates TEM image of FMPSi-Cis@GO@DQA.

### 3.2. In Vitro Drug Release under pH/NIR Irradiation

#### 3.2.1. Photothermal Heating by NIR Irradiation

To investigate the photothermal heating effect on silicon-based nanocarriers, as-prepared samples were irradiated with NIR light (808 nm, 1.0 W/cm^2^) for 20 min, 22 cm of NIR irradiation distance, and 12 J/cm^2^ of energy density. According to Figure 2A, the temperatures of pure PBS and MSN solutions (4 mg/mL) were increased from 23.7 °C to 33.1 °C and 34.5 °C, respectively. On the other hand, the temperature of MPSi solution was more significantly increased from 23.7 °C to 45.3 °C, and the GO-wrapped FMPSi exhibited the maximal increase in temperature (up to 52.8 °C). These results indicate that silicon nanostructure with numerous mesopores exhibits high photo-induced hyperthermia due to its excellent photothermal conversion efficiency, high surface area, and low reflectivity (or antireflection) [64,65]. GO wrapping can further increase NIR absorption through the laser-induced reduction of GO [66,67].

#### 3.2.2. In Vitro Release Test by pH Changes

The FMPSi-Cis@GO was dissolved in 10 mL of PBS under constant stirring at 37 ± 1 °C. During in vitro release test, the solution was periodically sampled to measure the absorbance changes at 301 nm by UV–Vis spectroscopy (NanoDrop; NanoDrop Technologies, Wilmington, DE, USA). The calculated absorbance changes were converted as the released amounts of Cis using the standard curve shown in Appendix A. According to Figure 2B, the cumulative release fraction was 13.9% (at pH 7.4) and 29.6% (at pH 5.5) at 10 h. After that, the release fraction was increased to 21.1% (at pH 7.4) and 39.6% (at pH 5.5) at 30 h, and slowly approached an asymptotic value of 24.8% (at pH 7.4) and 50.1% (at pH 5.5) after 60 h. At acidic pH 5.5, the carboxylic acid groups are prone to be protonated, possibly leading to the aggregation of GO nanosheets with less hydrophilicity [68]. The self-aggregation of GO weakens the intermolecular interactions of GO nanosheets with the FMPSi, resulting in the fast drug release [36,69]. At neutral pH 7.4, however, the deprotonated carboxyl acid groups are more hydrophilic, and the GO nanosheets tend to have stronger interactions with the FMPSi, resulting in the slow drug release [70]. As a result, the FMPSi-Cis@GO showed the larger difference of release fractions between pH 5.5 and pH 7.4, as shown in Figure 2B.

#### 3.2.3. In Vitro Release Test under NIR Irradiation

To investigate the dual stimuli (pH and NIR irradiation)-responsiveness of drug release behavior, in vitro release tests of FMPSi-Cis@GO were carried out in PBS under NIR irradiation (808 nm, 1.0 W/cm^2^) at different pHs (at pH 5.5 and 7.4). The NIR laser was periodically irradiated for 20 min. During the in vitro release process, the solution was periodically sampled, and the absorbance was measured at 301 nm using UV–vis spectroscopy. As shown in Figure 2C, the release rate of FMPSi-Cis@GO (pH 7.4 and 5.5) was increased significantly under NIR light irradiation (808 nm, 1.0 W/cm^2^) for 20 min. The short violet lines indicate the temperature changes of the pure solution by periodic NIR irradiations at 0, 5, 10, 20, and 30 h.

For FMPSi-Cis@GO at pH 7.4, the solution temperature increased to the maximal value of 52.8 °C under the NIR irradiation, followed by the decrease in solution temperature to 38.3 °C within 60 min (1 h) after removing the NIR laser. After the first NIR irradiation for 20 min, the release fraction of FMPSi-Cis@GO reached 12.1%. After periodic NIR irradiations, the release fraction increased drastically from 18.8% to 29.4% (second NIR irradiation at 5 h), 32.1% to 43.0% (third NIR irradiation at 10 h), 47.5% to 55.6% (fourth NIR irradiation at 20 h), and 60.9% to 66.4% (fifth NIR irradiation at 30 h). The release fraction finally reached an asymptotic value of 75.7% at 60 h. In contrast, the release fraction of FMPSi-Cis@GO without NIR irradiation slowly approached an asymptotic value of 24.8% at 60 h.

At acidic pH 5.5, the release fraction of FMPSi-Cis@GO more rapidly reached 19.8% by the first NIR irradiation, which continuously increased from 32.6% to 49.2% by the second NIR irradiation, from 57.2% to 63.6% by the third NIR irradiation, from 69.5% to 72.8% by the fourth NIR irradiation, and finally reached to 77.9% by the fifth NIR irradiation. The release fraction gradually approached an asymptotic value of 79.2% after 40 h.

### 3.3. Release Kinetics and Stimuli-Responsive Mechanisms

#### 3.3.1. Comparative Release Kinetics

The release kinetics of Cis from pure FMPSi was measured in PBS at pH 5.5 and pH 7.4, respectively. According to Figure 2D, the initial burst of release was attributed to the rapid dissolution of Cis located near the surface. The prolonged-release behavior stems from the Cis entrapped inside the porous channels of FMPSi [8]. According to Appendix A (Appendix A) for the Fickian diffusion model, the FMPSi-Cis showed k_F_ = 0.68 and 0.44 at pH 7.4 and 5.5, respectively. The difference of these k_F_ values indicated that the release rate of Cis at pH 5.5 was lower than that at pH 7.4, probably due to the slight difference of the chemical structure of Cis at different pH conditions. At acidic pH 5.5, the hydrolysis of *cis*-PtCl_2_(NH_3_)_2_ leads to the formation of various forms of *cis*-PtCl(NH_3_)_2_(OH_2_)^+^ and *cis*-Pt(OH)(NH_3_)(OH_2_)^+^ [71,72]. The surface of FMPSi contains polar functional groups (such as OH, Si-O, and COO) that can participate in polar and electrostatic interactions with the charged Cis complexes. As a result, Cis drug tends to be released slowly through the porous channels at acidic pH 5.5 than at neutral pH 7.4 [9]. The release kinetics of FMPSi-Cis@GO at pH 5.5 were analyzed by Appendix A for a power–law diffusion model, which produced the fitted values of *k_R_ =* 12 and *n* = 0.69, indicative of non-Fickian diffusion caused by the retarded transmission of Cis through the GO layer wrapped over the FMPSi. All fitted parameter values were summarized in Appendix A.

#### 3.3.2. Stimuli (pH and NIR)-Responsive Drug Release Mechanisms

The drug release mechanisms of multifunctional MPSi are strongly dependent on pH conditions, as shown in Figure 2E. At neutral pH 7.4, the release rate of FMPSi-Cis@GO was significantly retarded due to the blocking layer of GO nanosheets. However, the release rate of FMPSi-Cis@GO at pH 5.5 was increased significantly, indicating the disruption of GO nanosheets on the FMPSi. The increase in protonated carboxylic acid groups at acidic pH 5.5 induced the self-agglomeration of GO nanosheets, consequently leading to the detachment of GO nanosheets from the FMPSi [73,74]. Meanwhile, the FMPSi-Cis@GO under NIR irradiation exhibited a significantly enhanced release rate at pH 5.5 as compared to that at pH 7.4, as shown in Figure 2F. GO nanosheets are known to absorb NIR light efficiently due to the delocalization of electrons across all adjacent π bonds in the GO. As a result, GO wrapping over the NPs can be weakened by the high thermal energy converted from absorbed NIR light [68], which is more aggravated by the disruption of GO nanosheets at acidic pH [75]. Our multifunctional NPs can be a prospective candidate for chemo-photothermal nanocarrier against cancer cells, because they can provide dual stimuli (pH and NIR)-responsive controlled drug release.

In summary, GO-wrapped FMPSi (FMPSi-Cis@GO) exhibited the highest release rate under NIR irradiation at pH 5.5. The photothermal synergy effect was attributed to high NIR absorption and self-aggregation of GO at acidic pH. The release of Cis from FMPSi-Cis@GO was facilitated by increasing the ratio of deprotonated to protonated carboxylic acid groups in GO, i.e., the protonated GO is less hydrophilic and more self-agglomeration state, leading to a detachment of GO from the FMPSi [61]. Thus, under NIR irradiation, the release rates of FMPSi-Cis@GO reached 77.9% at pH 5.5 and 75.7% at pH 7.4, due to the high photothermal conversion efficiency of GO [76].

### 3.4. Cell Targeting, Toxicity, and Mitochondrial Uptake

#### 3.4.1. Confocal Microscopy for Mitochondria Targeting

For the cell experiments, the three cell lines HEK-293 (human embryonic kidney cells), HeLa (human cervical cancer), and SY-SY5Y (human neuroblastoma) were utilized. Cell experiments still proved to be the most economical and ethically viable method for performing scientific research. The HEK-293 cell is widely used as standard for normal human cells, whereas HeLa is a commonly used cancer cell line. Additionally, the SH-SY5Y cell was also used as the research model for human neuronal tumors, and this is also commonly used for toxicology evaluation.

To evaluate the cell targeting of the prepared NPs (MPSi, FMPSi, FMPSi-Cis@GO, and FMPSi-Cis@GO@DQA), HeLa cells were incubated with the NPs (10 μg/mL) for 4 h, 8 h, and 12 h, followed by treatment with Mito-Tracker Red. Mitotracker is a red fluorescent dye that explicitly stains the mitochondria. Hence, the co-localization by both the NPs (green fluorescence) and mitotracker (red fluorescence) can yield a yellow fluorescence as the merged image. In Figure 3, clean green spots were attributed to the fluorescence of FITC-labeled NPs, demonstrating the successful intracellular uptake of NPs. Importantly, a higher intensity of FITC fluorescence was obtained from DQA-conjugated NPs (FMPSi-Cis@GO@DQA) during the observation period. As shown in Figure 3A, green fluorescence indicated the location of FITC-labeled NPs in the tested cells. Mito-Tracker Red was used to track red-labeled mitochondria within live cells utilizing the mitochondrial membrane potential. The yellow dots in the merged image of green and red fluorescence (Figure 3A—right) revealed that the FMPSi-Cis@GO@DQA were mostly accumulated around the cell nucleus, demonstrating the successful localization of NPs and their targeting effectiveness to mitochondria. In the merged images, the yellow dots in the cytoplasm of the cells treated with FMPSi-Cis@GO (Figure 3A—left) were less distinct than that of the FMPSi-Cis@GO@DQA. The results demonstrated the efficient mitochondria–targeting of DQA conjugated NPs in the HeLa cancer cells. A similar experiment was carried out using FMPSi (Appendix A) and FMPSi-Cis samples without DQA conjugation (Appendix A). However, neither samples showed the yellow fluorescence in the merged images, indicating that DQA-free NPs were not effective for targeting the mitochondria.

#### 3.4.2. Cytotoxicity Assay

To investigate the anticancer activity of Cis-loaded NPs and the cytotoxicity of NP-based delivery platforms, the cytotoxicity assay was performed using HeLa and SH-SY5Y cancer cell lines, including normal HEK293 kidney cells (Figure 3B). In the Cell Titer-Glo^®^ Luminescence assay, the higher intensity in the cytotoxicity assay indicates the cell survival ratio. In both HeLa and SH-SY5Y cancerous cells, more than 80–85% cells were alive after being treated with MPSi (IC_50_ > 100 μg/mL) and FMPSi (IC_50_ > 100 μg/mL) at high loading of NPs (50–100 μg/mL) for 48 h incubation, indicating good biocompatibility and no obvious cytotoxicity of the NPs in the treatment of cancerous cells without drug loading. FMPSi exhibited a slightly higher cell survival rate than that of MPSi in the concentration range of 6.5–12.5 μg/mL.

In the case of SH-SY5Y cells, FMPSi-Cis@GO (IC_50_ = 31.2 μg/mL) induced 24.1% cell death at 6.5 µg/mL loading, but FMPSi-Cis@GO@DQA (IC_50_ = 29.2 μg/mL) increased the rate of cell death up to 37.8% for the same loading amount due to the combination of anticancer activity of Cis drug and mitochondrial targeting agent of DQA [77].

Utilizing the HeLa cells, FMPSi-Cis@GO@DQA (IC_50_ = 24.4 μg/mL) also exhibited a higher cytotoxicity compared to FMPSi-Cis@GO (IC_50_ = 31.6 μg/mL) at 12.5–25 μg/mL concentrations. The HeLa cells almost showed anti-proliferation at 50 μg/mL concentration, corroborating to the observed FITC fluorescence as shown in Figure 3A (right). In addition, most HEK293 cells survived over the whole concentration range for all the NPs (IC_50_ > 100 μg/mL), indicating no obvious cytotoxicity of NPs against normal cells, except for the highest concentration of 100 µg/mL. These results indicate that the synthesized NP can potentially prevent the in vitro proliferation of the cancer cell lines HeLa and SH-SY5Y but are not harmful to normal HEK cells (IC_50_ > 100 μg/mL). Recently, high concentrations of silicon NPs (at 30 and 100 μg/mL) have been reported to have a cytotoxic effect on HEK293 cells [78]. In contrast, FMPSi-based NPs in our study did not show obvious cytotoxicity to normal cells.

We also investigated the cytotoxic effects of DQA on the three cell lines. As the amount of DQA used in the preparation of the NP is at a maximum of 5%, 0–5 μg/mL concentrations were used for the cytotoxicity assay. Interestingly, DQA only exhibited a minimal and non-significant cytotoxic effects at 5 μg/mL (% inhibition < 50%) (Appendix A). Hence, the observed cytotoxicity of the synthesized NP could be attributed to the synergistic effects of the different ligands.

#### 3.4.3. Mitochondria Isolation and Analysis of Fluorescence Intensity

To track the mitochondria binding capability of the NPs (FMPSi-Cis@GO and FMPSi-Cis@GO@DQA), the FITC fluorescence was detected in isolated mitochondria from the treated HeLa, SH-SY5Y, and HEK293 cells. The mitochondria of various cancer cells treated with FMPSi-Cis@GO and FMPSi-Cis@GO@DQA were isolated using the mitochondria isolation kit (Thermo Scientific). As shown in Figure 4A, FITC fluorescence intensities of isolated mitochondria extracted from humane cells showed distinct differences between the normal cells (control) and cancerous cells. After being treated with FMPSi-based NPs (FMPSi-Cis@GO and FMPSi-Cis@GO@DQA), the FITC fluorescence intensity in the isolated mitochondria from the cancer cell lines (HeLa and SH-SY5Y) was significantly higher than that from the normal HEK293 cells. In addition, measurement of the FITC fluorescence intensities of the mitochondria fraction isolated from cancerous cells showed that the FMPSi-Cis@GO@DQA-treated subjects exhibited 50% higher fluorescent intensity than that of FMPSi-Cis@GO-treated subjects.

As an early event in mitochondria triggered apoptosis, the loss of mitochondrial membrane potential (Δψm) is closely related to mtDNA expression alterations. To assess mitochondrial dysfunction induced by the NPs, we measured the changes of intracellular mitochondrial membrane potentials by treating FMPSi-Cis@GO and FMPSi-Cis@GO@DQA for a short time of 24 h. TMRE mitochondrial membrane potential assay kit (Abcam) was used to measure the mitochondrial potentials of cancerous HeLa and SH-SH5Y cells. Then, FCCP was used as a positive control in this KIT, and fluorescence intensity was used for assessing mitochondrial membrane potentials (Δψm) [79]. As shown in Figure 4B, the mitochondrial membrane potentials of cells treated with the NPs (FMPSi-Cis@GO and FMPSi-Cis@GO@DQA) were significantly decreased compared to those of the negative control group, and there was no noticeable difference of fluorescence intensity between in HeLa and SH-SY5Y cells apoptosis. This result indicated that the mitochondrial inner membranes were damaged by the short-term treatment (24 h) with FMPSi-based NPs.
Figure 4The rationale for mitochondria-targeting of multifunctional mesoporous silicon nanoparticle system to assess Δψm. (**A**) FITC fluorescence absorption intensities of isolated mitochondria measured from HeLa, SH-SY5Y, and HEK293 cells without and with the indicated treatments; (**B**) Mitochondrial membrane potential of HeLa and SH-SY5Y cells after treatment with FMPSi-Cis@GO and FMPSi-Cis@GO@DQA for a short time of 24 h incubation. PBS treatment was the negative (control) group; FCCP treatment was a positive control. Δψm was measured by fluorescence intensity after cells were stained with TMRE. Data are represented as the mean ± SD (*n* = 3); and (**C**) Schematic illustration of specific mitochondrion targeting and therapeutic effects of multifunctional mesoporous silicon nanoparticle system.
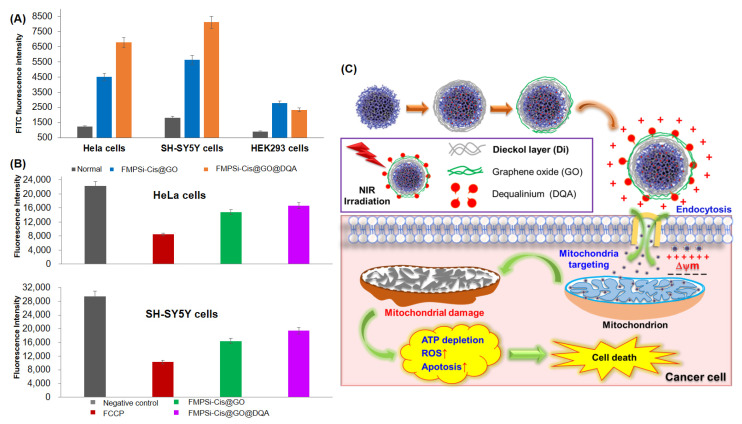


Owing to the amphiphilic nature of DQA, it can be self-assembled as liposome-like cationic vesicles, taking advantage of the highly negative Δψm in cancer cells [80]. That is, DQA-conjugated NPs can be an effective mitochondriotropic carrier that delivers cytotoxic drugs into cancerous cells by activating nucleases in the inner membrane and the matrix of mitochondria [81,82]. That is, the DQA-conjugated NPs can selectively accumulate in mitochondria by reducing the membrane potential and damaging mitochondrial membrane, finally activating reactive oxygen species (ROS) production, which can inhibit the production of ATP and induce apoptotic cell death (Figure 4C). Hence, it is tentatively concluded that DQA-conjugated NPs (FMPSi-Cis@GO@DQA) have the excellent capability of targeting the mitochondria of cancerous cells, as well as equipping them with stimuli-responsive controlled drug release.

To obtain direct evidence for NPs internalization into human cells, HeLa cancer cells were co-cultured with 50 μg/mL of FMPSi-Cis@GO and FMPSi-Cis@GO@DQA for 24 h. Subsequently, the cross-sectioned cells were analyzed using a low-resolution TEM [83]. As shown in the left panels of Figure 5, the localization of aggregated NPs (FMPSi-Cis@GO and FMPSi-Cis@GO@DQA) were observed in specific areas in the cell (Figure 5c,e). On the other hand, control NPs (MPSi) were dispersed in the whole cytoplasmic area in the HeLa cell (Figure 5a). As shown in the right panels of Figure 5, the magnified TEM images indicated the accumulation of the FMPSi-based NPs in the mitochondria region of cancer cells (Figure 5d,f). Furthermore, changes in cell morphology and damages to the mitochondrial structure were induced by intracellular NPs. Structural changes in cell membranes (such as mitochondrial swelling and cristae rupturing) are observed in the left panels of Figure 5c,e. In summary, the NPs entered the cells through different pathways and were dispersed in the cytoplasm and accumulated inside mitochondria. The delivered cytotoxic NPs consequently led to mitochondrial damages, which were caused by oxidative stress and/or direct injurious effect of the NPs [52,53].

## 4. Conclusions

In this work, silicon-based nanocarriers were developed for the specific mitochondria-targeting in cancer cells, as well as being designedwith dual stimuli-responsive controlled drug release. Specifically, mesoporous silicon NPs were prepared via magnesium reduction of mesoporous silica NPs, which was subsequently functionalized with internal fluorescent conjugates and external graphene oxide nanosheets, so-called fluorescent mesoporous silicon (FMPSi) NPs. Notably, GO-wrapped FMPSi (FMPSi@GO) exhibited the dual stimuli (pH and NIR)-responsive drug release behaviors, i.e., the release rate at pH 5.5 was higher than that at pH 7.4 and significantly enhanced under NIR irradiation at acidic pH 5.5. Furthermore, DQA-conjugated FMPSi@GO NPs demonstrated an excellent mitochondrial targeting specificity and highly selective accumulation in the cancer cells, leading to apoptotic death of cancer cells while there was no noticeable toxicity to normal cells. Our innovative silicon nanocarriers displayed not only regulated drug release in response to dual stimuli (pH and NIR), but also targeted accumulation and destruction of cancer cell mitochondria. This research contributes to develop the advancement of cancer treatment in a more promising, effective, and rapid way.

## Data Availability

Not applicable.

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
