# Peer review of "Dual Stimuli-Responsive Multifunctional Silicon Nanocarriers for Specifically Targeting Mitochondria in Human Cancer Cells"

_pharmaceutics, 2022, doi:10.3390/pharmaceutics14040858_

Round 1

Reviewer 1 Report

SIGNIFICANCE OF THE WORK:

In this paper fluorescent silicon nanoparticles for the delivery of cys-Pt have been prepared. The particles were coated with graphene oxide (GO). The design of such coating endows the system with pH and temperature sensitivity. The surface of the particles has been decorated with dequalinium for mitochondrial targeting.

METHODOLOGY:

The prepared materials have been thoroughly characterized by typical methods.

TEXT:

The manuscript is well-written, and in general the figures are well crafted and informative. However, Figure 2 is quite difficult to read.

NOVELTY

Many reports deal with the use of GO for MSN gating (cf J. Mater. Chem. B, 2015, 3, 6377-6384). However, the present system is, to the best of my knowledge, new.

COMMENTS:

Although the design and synthesis of the system is quite interesting, some points should be addressed before publication:

  1. Hemocompatibility of FMPSi-Cis@GO@DQA and protein corona formation should be carefully studied.
  2. The release characterization has been only carried out for FMPSi@GO. In my opinion, these studies should include the complete system FMPSi-Cis@GO@DQA.
  3. Better TEM images for systems FMPSi-Cis@GO and FMPSi-Cis@GO@DQA should be provided (Figure 1 D).
  4. Confocal images should show with more detail the colocalization of the particles with mitocondia.

Author Response

Manuscript ID: pharmaceutics-1660947

Title: Dual Stimuli–Responsive Multifunctional Silicon Nanocarriers for

Specifically Targeting Mitochondria in Human Cancer Cells

RESPONSE TO THE REVIEWER

We would like to thank for the comments of the editor and reviewers through 3 rounds to improve the manuscript significantly. We have spent a lot of time and careful review of the entire manuscript to correct and supplement the comments of editors and reviewers. These changes have been carefully and appropriately added in the revised manuscript file or supplementary material file. In addition, we respond to the Reviewer's comments below, point by point, and describe the associated revisions to the manuscript.

With efforts and try to improve this research during the past time. We hope that this revised manuscript will satisfy the requirements of the editor and the journal. We confirm that the manuscript nor any parts of its has not been submitted or published in another journal. All authors have approved the manuscript and agree with its submission to Pharmaceutics.

Please see the attached files.

Sincerely yours,

Reviewer 2 Report

The article entitled Dual Stimuli–Responsive Multifunctional Silicon Nanocarriers for Specifically Targeting Mitochondria in Human Cancer Cells The authors dequalinium (DQA)-conjugated FMPSi@GO, FMPSi@GO@DQA, provided the nanocarrier with re- markable mitochondria targeting specificity, consequently leading to a significant de- crease in cellular ATP production and damaging mitochondrial membranes of cancer cells. In my opinion, the article sounds interesting but requires revisions prior to being publishable.

  1. I suggest authors explain briefly the rationale for the use of HeLa, SH-SY5Y, and HEK293 cell lines.
  2. MSNs are different to silicon nanoparticles, I suggest checking the writing carefully,
  3. Better to mention NIR irradiation distance? Also, energy density?
  4. Require to provide drug loading and drug efficient percentage?
  5. Very poor editing of the manuscript. Yellow highlights, irregular justification of the text, different line spacing in different paragraphs.
  6. All provided abbreviations should be checked.
  7. Suggest polishing the language as well.

Author Response

(The authors gave the same response as above.)

Round 2

Reviewer 1 Report

Thank you very much for the revised manuscript. The authors have answer my questions and comments.

This manuscript is a resubmission of an earlier submission. The following is a list of the peer review reports and author responses from that submission.

Round 1

Reviewer 1 Report

The manuscript entitled “Dual Stimuli–Responsive Multifunctional Silicon Nanocarriers for Specifically Targeting Mitochondria in Human Cancer Cells” reported the development of silicon-based nanocarriers for the specific mitochondrial targeting in cancer cells with dual stimuli (pH, NIR)-responsive drug release behaviors. The manuscript is well written and presented and it is suitable for publication in Pharmaceutics.

My comments are as follows:

- Please provide the yield of the reaction at each synthesis steps (from the synthesis of mesoporous silica nanoparticles to wrapping with GO layers, and conjugation with QDA) as amount of nanoparticles recovered and degree of functionalization.

- Cytotoxicity study: FMPSi@GO, and FMPSi@GO@DQA (without Cys) should be added as control. From the statistical point of view, the differences between different NPs should be determined based on IC50 (the estimated NP concentration killing 50% of the cells).  IC50 should be calculated for each NP type.

- It is  beneficial to mention more in detail the application as type of cancer and which is the desired administration route. It will be important to supplement more detailed discussion or prospect of future study.

- Please check the Pharmaceutics layout

Author Response

February 12, 2022       

Manuscript ID: pharmaceutics-1582918

Title: Dual Stimuli–Responsive Multifunctional Silicon Nanocarriers for

Specifically Targeting Mitochondria in Human Cancer Cells

RESPONSE TO THE REVIEWER 1

We are grateful to the Reviewer 1 for a careful and thorough review, and for raising important issues that relate directly to the clarity of the manuscript and the interpretation of the data. We responded to the Reviewer’s comments below, and described the associated revisions to the manuscript. We hope that our revision merits acceptance of our manuscript.

The manuscript entitled “Dual Stimuli–Responsive Multifunctional Silicon Nanocarriers for Specifically Targeting Mitochondria in Human Cancer Cells” reported the development of silicon-based nanocarriers for the specific mitochondrial targeting in cancer cells with dual stimuli (pH, NIR)-responsive drug release behaviors. The manuscript is well written and presented and it is suitable for publication in Pharmaceutics.

My comments are as follows:

  1. Please provide the yield of the reaction at each synthesis steps (from the synthesis of mesoporous silica nanoparticles to wrapping with GO layers, and conjugation with QDA) as amount of nanoparticles recovered and degree of functionalization.

Authors Response:

Thank you for your constructive question, we have tried our best to double check the experiment, and calculated the yield of each synthetic step as follows. We hope that these supplements will meet your requirements.

  • Synthesis of silica nanoparticle (MSN)

The yield of the SiO2 was calculated from the weight of the collected after the steps of synthesis versus the theoretical weight of SiO2 obtained from TEOS. Composite diagram as shown below:

TEOs (Molar mass: 208.33 g⋅mol−1, Density: 0.933 g/mL at 20 °C),  

SiO2 (Molar mass: 60.084 g⋅mol−1).

Theoretically, 10 ml of TEOs will produce 2.69 mg of SiO2. In practice, SiO2 was synthesized in section 1.2. Synthesis of nanocarriers has an efficiency of 71.4% (1.92mg/2.96mg).

  • Synthesis of silicon nanoparticle (MSN into MPSi)

The process of synthesizing Si from SiO2 by magnesium is shown as shown below.

Theoretically 600 mg of SiO2 will produce 280 mg of Si. Actual experimental performance achieved 48.5% (136 mg/280 mg).

  • Wrapped with GO layers, and conjugation with QDA:

For the process GO layer covers the silicon bead. As described in the manuscript "GO (0.1 g; 0.025 mL of GO suspension) was sonicated in 2 mL of DI water for 3 min. The FMPSi-Cis (50 mg) and GO solution were mixed and dispersed in DI water using ultrasonic vibration for 3 min. After ultrasonic dispersion, the mixed solution was stirred at 60 oC for 2 h. The precipitate was washed with DI water several times. Then, the final product (FMPSi-Cis@GO) was collected by centrifugation and dried in the freeze dryer system."

We have experimentally performed to achieve a GO coating outside the MPSi nanoparticle as shown in the TEM image of Figure 1D. Based on the mass ratio of GO: MPSi which is 1:50 and the number of washes (specifically 3 times) we can control the outer edema layer of GO. This mass ratio is optimal for coating GO around silicon nanoparticles, and the right number of washes to give the right coating thickness.

For the conjugation step with DQA agent, we also optimized the holding ratio of DQA, EDC and NHS to achieve particle structures with DQA covering the GO nanosheets. The importance of DQA is to create a layer of positive charge sufficient to target mitochondria. Unfortunately, we did not take this into account during our implementation in this study.

Thank you for your input, we will note and implement in the next research. We hope that our explanations will satisfy your requirements. We have provided the yield of the reaction at each step in Supplementary Materials.

  1. Cytotoxicity study: FMPSi@GO, and FMPSi@GO@DQA (without Cys) should be added as control. From the statistical point of view, the differences between different NPs should be determined based on IC50 (the estimated NP concentration killing 50% of the cells). IC50 should be calculated for each NP type.

Authors Response:

Thank you for this valuable comment. We completely agree on the concern of cytotoxicity of NPs without Cis. Unfortunately, we were not able to determine the cytotoxicity of NP without Cis. However, we are in the process of investigating these NPs and we will take note of this comment in the future.

We have calculated the IC50 values of the NP and mentioned them on pages 12 and 15 of manuscript.

The synthesized NP FMPSi-Cis@GO@DQA proved to be efficient in inhibiting the growth of two cancer cell lines SH-SY5Y (IC50 = 29.2 mg/mL) and HeLa (IC50 = 24.4 mg/mL), while non-toxic to the normal HEK-293 cells (IC50 > 100 mg/mL). Currently, we are investigating the cytotoxicity of the NP to a variety of cell lines to properly document the most efficient targets (excellent IC50 value) of the NP. Further, other NPs with different or additional ligands are being undertaken. Collective NPs with outstanding activity will be subjected to 3D cell cytotoxicity and in vivo assays.

  1. It is beneficial to mention more in detail the application as type of cancer and which is the desired administration route. It will be important to supplement more detailed discussion or prospect of future study.

Authors Response:

We have added an additional statement on this comment on page 15 of manuscript.

The synthesized NP FMPSi-Cis@GO@DQA proved to be efficient in inhibiting the growth of two cancer cell lines SH-SY5Y (IC50 = 29.2 mg/mL) and HeLa (IC50 = 24.4 mg/mL), while non-toxic to the normal HEK-293 cells (IC50 > 100 mg/mL). Currently, we are investigating the cytotoxicity of the NP to a variety of cell lines to properly document the most efficient targets (excellent IC50 value) of the NP. Further, other NPs with different or additional ligands are being undertaken. Collective NPs with outstanding activity will be subjected to 3D cell cytotoxicity and in vivo assays.

  1. Please check the Pharmaceutics layout

Authors Response:

We've double-checked the entire article for layout as well as grammar and English errors. Thank you very much for your valuable comments.

Reviewer 2 Report

 The work presented covers the synthesis of fluorescein-functionalised silica nanoparticles coated with graphene for study as antitumour agents. The ability of these nanoparticles to release in a controlled manner Mitochondrial specific agents, namely Dequalinium by dual stimuli (ph and NIR).

The nanoparticles presented are well characterised and the studies of exposure to stimuli for agent release are well described and show promising results that are of interest to the scientific community so I recommend the publication of the manuscript in pharmaceuticals.

Author Response

February 12, 2022       

Manuscript ID: pharmaceutics-1582918

Title: Dual Stimuli–Responsive Multifunctional Silicon Nanocarriers for

Specifically Targeting Mitochondria in Human Cancer Cells

RESPONSE TO THE REVIEWER 2

We are grateful to the Reviewer 2 for a careful and thorough review, and for raising important issues that relate directly to the clarity of the manuscript and the interpretation of the data. After discussing his (her) report, we agree with the suggestion of the Reviewer 2 to submit the revised manuscript to Pharmaceutics.

The work presented covers the synthesis of fluorescein-functionalised silica nanoparticles coated with graphene for study as antitumour agents. The ability of these nanoparticles to release in a controlled manner Mitochondrial specific agents, namely Dequalinium by dual stimuli (ph and NIR).

The nanoparticles presented are well characterised and the studies of exposure to stimuli for agent release are well described and show promising results that are of interest to the scientific community so I recommend the publication of the manuscript in pharmaceuticals.

Authors Response:

We sincerely appreciate the positive comments and contributions to this manuscript. We hope that this study will be of wide interest to readers.

Reviewer 3 Report

  1. Figure 3a does not demonstrate a clear mitochondrial morphology through the red channel. For colocalization analysis with mitochondrial staining dyes, it is strongly recommended to provide images with high resolution and colocalization quantification (e.g. Pearson's correlation coefficient).
  2. The authors compared the toxicity of the formulation against three different cell lines and concluded that the material is cancer targeting. HEK293 cells are more or less a type of cancerous cell too. One possibility to cause the difference in cytotoxicity may be attributed to the sensitivity of the cells against mitochondrial targeting compounds. 
  3. One of the most critical control experiment is missing. As dequalinium itself is a delocalized lipophilic cation, it is already a mitochondrial targeting compound. The authors should provide the compound only group in all the toxicity and microscopy experiments. 

Author Response

February 12, 2022       

Manuscript ID: pharmaceutics-1582918

Title: Dual Stimuli–Responsive Multifunctional Silicon Nanocarriers for

Specifically Targeting Mitochondria in Human Cancer Cells

RESPONSE TO THE REVIEWER 3

We are grateful to the Reviewer 3 for a careful and thorough review, and for raising important issues that relate directly to the clarity of the manuscript and the interpretation of the data. We respond to the Reviewer’s comments below, point by point, and describe the associated revisions to the manuscript.

  1. Figure 3a does not demonstrate a clear mitochondrial morphology through the red channel. For colocalization analysis with mitochondrial staining dyes, it is strongly recommended to provide images with high resolution and colocalization quantification (e.g. Pearson's correlation coefficient).

Authors Response:

Thank you for the comment. We have changed Figure 3A with larger magnifications and high resolution in the revised version of manuscript.

Confocal images were re-taken with larger magnifications and high resolution. FITC and MITO showed discrete pixels around the nucleus. The co-localization by both the NPs (green fluorescence) and mitotracker (red fluorescence) yields a yellow fluorescence as the merged image.

  1. The authors compared the toxicity of the formulation against three different cell lines and concluded that the material is cancer targeting. HEK293 cells are more or less a type of cancerous cell too. One possibility to cause the difference in cytotoxicity may be attributed to the sensitivity of the cells against mitochondrial targeting compounds.

Authors Response:

Thank you for the comment. In our study, non-cancerous cell line derived from human embryonic kidney cells (HEK293; ATCC no. CRL-1573) was used as the control. This HEK293 cells are one of the most common cell lines widely used for cancer research as a representative model of normal human cells ideally. Repeatedly, numerous in vitro studies have been used these cells as a non-cancerous cell line to demonstrate biological activities of their synthetic NPs ​[Joseph et al., 2022; Liu et al., 2021; Benyettoua et al., 2017]

References

Joseph C, Daniels A, Singh S, Singh M. Histidine-Tagged Folate-Targeted Gold Nanoparticles for Enhanced Transgene Expression in Breast Cancer Cells In Vitro. Pharmaceutics. 2022; 14(1):53. https://doi.org/10.3390/pharmaceutics14010053

Liu X, Shan K, Shao X, et al. Nanotoxic Effects of Silver Nanoparticles on Normal HEK-293 Cells in Comparison to Cancerous HeLa Cell Line. Int J Nanomedicine. 2021;16:753-761. Published 2021 Feb 3. doi:10.2147/IJN.S289008

Benyettoua F., H. Fahsa, R. Elkharraga, R. A. Bilbeisia, B. Asmaa, R. Rezguia, L. MotteORCID logob, M. Magzouba, J. Brandelc, J.-C. Olsend, F. Pianoae, K. C. Gunsalusae, C. Platas-Iglesiasf and A. Trabolsi. Selective growth inhibition of cancer cells with doxorubicin-loaded CB[7]-modified iron-oxide nanoparticles. RSC Adv., 2017, 7, 23827-23834

  1. One of the most critical control experiment is missing. As dequalinium itself is a delocalized lipophilic cation, it is already a mitochondrial targeting compound. The authors should provide the compound only group in all the toxicity and microscopy experiments.

Authors Response: Thanks for your comment on dequalinium (DQA). DQA plays a role in targeting cancer cell mitochondria from nanocarriers carrying active ingredients to kill cancer cells. As included in the references (Bae et al., 2018; Wang et al., 2011), DQA was already to target mitochondria in conjunction with other functional groups on nanoparticles (e.g. DQA80s, DQA-PEG(2000)-DSPE). Thus, we took the next step from the previous reports and applied it to our nanoparticle. Herein, to enhance mitochondrial selectivity and cancer cell toxicity, DQA was further conjugated with FMPSi-Cis@GO to develop mitochondria-targeting nanocarriers as FMPSi-Cis@GO@DQA, which could transport chemotherapeutics to mammalian mitochondria in living cancer cells. It can be said that DQA allows compounds to preferentially enter the mitochondria of cancer cells. The mitochondria of cancer cells usually have a higher membrane potential compared to those of normal cells, which enables DQA to selectively target cancer cells and accumulate more rapidly in their mitochondria. A previous study has also demonstrated that lipophilic cationic compounds accumulated and were retained in mitochondria of living cancer cells for significantly longer periods than in normal cells (Bleday et al., 1986). DQA can also have synergistic effects when used with other anti-cancer drugs (Bae et al., 2018). DQA and cisplatin have a good synergistic effect when combined with a ratio of 1:1 or 2:1, which greatly enhances the cytotoxicity of cisplatin to drug-resistant tumour cells (Cheng et al., 2018). Altogether recent publications indicate that DQA exerts a wide range of anti-cancer activity through its ability to target and accumulate in the mitochondria. Hence, DQA in combination with other drugs as cisplatin or the modification of nanoparticles of other chemotherapeutic drugs are promising, safe and economical means for anticancer therapy. We kindly agree on the concern of cytotoxicity of the NPs without Cis. Unfortunately, we did not determine the cytotoxicity of NPs without Cis since we did take the next step from the previous reports and applied it to our nanoparticle.  However, we will take note of this comment in the future. Thank you very much for your helpful comments.

References

Bae, Y.; et al. Dequalinium-based functional nanosomes show increased mitochondria targeting and anticancer effect. Eur. J. Pharm. Biopharm. 2018, 124, 104-115.

Wang, X.-X.; et. al. The use of mitochondrial targeting resveratrol liposomes modified with a dequalinium polyethylene glycol-distearoylphosphatidyl ethanolamine conjugate to induce apoptosis in resistant lung cancer cells. Biomaterials 2011, 32, 5673-5687.

Bleday R, Weiss MJ, Salem RR, Wilson RE, Chen LB, Steele G Jr. Inhibition of rat colon tumor isograft growth with dequalinium chloride. Arch Surg. 1986;121(11):1272-1275.

Bae Y, Jung MK, Lee S et al. Dequalinium-based functional nanosomes show increased mitochondria targeting and anticancer effect. Eur J Pharma Biopharm. 2018;124:104-115.

Cheng AN, Lo Y-K, Lin Y-S et al. Identification of novel Cdc7 kinase inhibitors as anti-cancer agents that target the interaction with Dbf4 by the fragment complementation and drug repositioning approach. EBioMedicine. 2018;36:241-251.

Round 2

Reviewer 3 Report

1. The authors did not fully address the comments from previous reviewers. The references to support their opinions were not to the point. For example, if the argument about whether the HEK cell line is a cancerous cell line, the supporting references should be from cell biology research (e.g. DOI: 10.1016/j.gene.2015.05.065). The reason why the reported system have different effects on three different cell lines should be included. 

2. It is critical to provide the result of the DQA only control experiment, regardless of previous reports - the comparison has to be done in parallel. 

3. Does Figure 3A indicate that mitochondria can also show up in the nucleus?